# Twenty Years of GH Treatment in Adults with Prader-Willi Syndrome

**DOI:** 10.3390/jcm10122667

**Published:** 2021-06-17

**Authors:** Anna Sjöström, Charlotte Höybye

**Affiliations:** 1Department of Clinical Chemistry, Karolinska University Hospital, 171 76 Stockholm, Sweden; anna.sjostrom@sll.se; 2Department of Molecular Medicine and Surgery, Karolinska Institutet, 171 76 Stockholm, Sweden; 3Department of Endocrinology, Karolinska University Hospital, 171 76 Stockholm, Sweden

**Keywords:** Prader-Willi syndrome, adults, GH treatment, long-term effects, long-term safety

## Abstract

Prader-Willi syndrome (PWS) is a rare neurodevelopmental genetic disorder. In adults, the syndrome is characterised by muscular hypotonia, a different body composition with more body fat than muscle mass, hyperphagia, behavioural problems, and cognitive dysfunction. Endocrine deficiencies are common, including growth hormone (GH) deficiency. Here, we present data from a cross-sectional study in adults with PWS with a focus on the long-term safety of GH treatment. A total of 22 patients (14 men) were treated with GH for a median of 20 years. Data on body composition, hormones, and metabolic parameters were retrieved from the patients’ medical records. The median age was 27 years. The median GH dose was 0.5 mg/day. Insulin-like growth factor 1 (IGF-I) and blood lipids were normal, while fasting glucose and HbA1c were slightly elevated in three men with diabetes. Fat mass was less than fat free mass in all, though this was less pronounced in women. GH treatment did not negatively affect the metabolic profile, and none developed cardiovascular diseases or cancer. All adults on long-term GH treatment had a normal body composition and our results indicate that treatment was safe. However, PWS is a complex, multisystemic disease and continuous, individual considerations are required during GH treatment, especially in patients with risk factors for adverse effects.

## 1. Introduction

Prader-Willi syndrome (PWS, OMIM 176270) is a rare neurodevelopmental genetic disorder with an incidence of about 1 in 15,000 children [1,2,3]. PWS is caused by the lack of expression of paternal genes in the q11–q13 region of chromosome 15 [1,2]. The genetic mechanisms include paternal microdeletion in 65–70%, maternal uniparental disomy in 30–35%, and imprinting defects in 1–3%. Characteristic symptoms are impaired cognitive function, muscular hypotonia, reduced exercise capacity, and an abnormal body composition with more body fat than lean body mass [4,5]. Endocrine deficiencies are common in PWS, including growth hormone (GH) deficiency (GHD), sex hormone deficiencies, hypothyroidism, and, in rare cases, hypocortisolism. Distinct nutritional stages represent the natural history of PWS [6]. Hyperphagia predominates in adults and requires strict environmental control and restricted access to food to prevent extreme obesity. Behavioural problems are often prominent and include an excessive interest in food, skin picking, repetitive behaviours, temper tantrums, anxiety, and mood instability [7]. Scoliosis and low bone mineral density are very frequent [8].

GH treatment has been approved for the treatment of children with PWS for many years and is still the only approved treatment for PWS. GH treatment in children with PWS has consistently been shown to improve growth, body composition, metabolism, cognitive development, and muscular function [9]. In contrast, GH treatment is not approved for the treatment of adults with PWS. In 2020, a systematic review of 20 GH trials in adults with PWS including data from 364 patients reported that all studies showed an improved body composition with an increase in lean body mass and a decrease in body fat (BF) [10]. The median duration of GH treatment in the reviewed studies was 2 years. In addition, two small observational studies of GH treatment lasting five and fifteen years both showed maintained, improved body composition [11,12]. In the systematic review from 2020, adverse events were reported in 7 of the 20 reviewed studies, including pretibial oedema, headache, and transient impaired glucose intolerance [10].

In summary, studies of adults with PWS have shown that GH treatment has beneficial effects on body composition and is safe. However, most previous studies had a relatively short duration, and knowledge of long-term outcomes and side effects is limited. In the present cross-sectional study, we report data with a focus on safety in a group of adults with PWS who had received long-term GH treatment.

## 2. Patients and Methods

Inclusion criteria was adult age (>18 years); genetically confirmed PWS diagnosis; a follow-up visit in the period 2017–2020 at the Department of Endocrinology, Karolinska University Hospital; and analyses of blood samples performed at the hospital’s laboratory. Data from the patients’ last visit at the Department of Endocrinology were retrieved from the patients’ medical records.

The total cohort was divided into two groups. One group had continuous treatment with GH either initiated in adulthood or in childhood with at least one visit one year following transition from the Paediatric Department. The other group consisted of non-GH treated adults naïve for GH treatment for more than one year.

The study was approved by the Swedish Ethical Authority (Dnr: 2020-06805).

### 2.1. Anthropometric Measurements and Body Composition

Physical examination included measurements of height and weight. Body mass index (BMI) was calculated as weight divided by the square of height in meters, kg/m^2^. A BMI from 18.5 to 25 kg/m^2^ was defined as normal, between 25 and 30 kg/m^2^ as overweight, and above 30 kg/m^2^ as obese according to the WHO criteria. FM and fat free body mass (FFM) were calculated and estimated from measurements of bioelectrical impedance (Tanita, Amsterdam, The Netherlands). The measurements were normalised for gender, age, and weight.

### 2.2. Endocrine and Metabolic Measurements

Blood samples for analyses of IGF-I; TSH; free T4 (fT4); free T3 (fT3); cortisol; fasting glucose; HbA1c; total, low-density (LDL), and high-density (HDL) lipoprotein cholesterol; and triglycerides were drawn fasting in the morning. All analyses were performed at the Clinical Chemistry Unit, Karolinska University Laboratory, using routine assays. In the Appendix A (Table A1), further information regarding analytes and measurements is displayed.

### 2.3. Statistics

Data are presented as medians and ranges. Differences were calculated using a Mann–Whitney U-test. Statistical significance was set at *p* < 0.05.

## 3. Results

A total of 33 adult patients with a genetically confirmed diagnosis of PWS were identified. Of these, 3 patients had UPD and 15 a deletion; 15 patients were methylation-positive, but detailed information of genetic subgroup was unknown. A total of 22 patients, 14 men and 8 women, were treated with GH, while 11 patients, 7 men and 4 women, were not. The median age of the total cohort was 32 (19–54) years. Three men, two in the GH treated group and one in the non-GH treated group, as well as two women in the non-GH treated group, were treated with insulin for type 2 diabetes. One man in the GH treated group was on oral anti-diabetics. One man in the GH-treated group and two men in the non-GH treated group were treated with statins. Five men in the GH treated group and one man in the non-GH treated group were on testosterone replacement.

Serum IGF-I was higher in the GH-treated group (*p* = 0.016), while TSH, fT4, fT3, and cortisol did not differ between the GH- and non-GH-treated groups (Table 1). Likewise, blood lipids were similar in the two groups (Table 1). HbA1c was higher in the non-GH-treated group (*p* = 0.024). No other differences in the metabolic profiles were noticed between genders.

### 3.1. The GH Treated Group

The median age was 27 years—34 years in men and 20 years in women (*p* = 0.0466). The median BMI was 27.4 kg/m^2^—27.0 kg/m^2^ in men and 29.5 kg/m^2^ in women. The median duration of GH treatment was 20 (10–35) years; 16 had been treated since early childhood and in 6 the treatment was initiated in adulthood. The median GH dose was 0.5 (0.2–1.2) mg. The highest GH doses were administered to the youngest individuals. All injections of GH were supervised by caregivers.

Body composition was compared between men and women. There were no differences between genders in BMI or FM (*p* = 0.407 and *p* = 0.124, respectively), whereas fat percentage and FFM differed (*p* = 0.003 and *p* = 0.0002, respectively) (Table 2). Three men were obese (highest BMI 37.5 kg/m^2^) and nine overweight. Three women were also obese (highest BMI 36.6 kg/m^2^) and three overweight. The fat percentage was less than 25% in five men, 25 to 30% in four men, and above 30% in five men. The fat percentage was above 30% in seven women and 26.2% in one woman. All patients had a higher FFM than FM (Figure 1). FFM was >10 kg higher than FM in all men, whereas the difference between FFM and FM was <10 kg in all women.

No complaints of oedema, muscle pain, or headache related to the GH treatment were reported. One man with diabetes had haemodialysis three times weekly. None of the GH-treated patients had been diagnosed with cancer or cardiovascular diseases.

### 3.2. The Non-GH Treated Group

Due to the small number of women (*n* = 4) in the non-GH treated group, the results for this group are summarised for all 11 patients. The median age in this group was 36 (19–48) years and the median BMI was 40.1 (20.5–53) kg/m^2^. Among the patients were two women and one man who continuously gained heavily in weight and developed type 2 diabetes. One woman developed uncontrolled diabetes at the age of 19 years, one woman at the age of 33 years, and the man at the age of 34 years. They had all been treated with GH since childhood. Over time, their diabetes became uncontrolled and GH treatment discontinued. Despite this, metabolic control was poor. Another five patients had never received GH treatment, one was treated for a short period during childhood, and two men discontinued GH treatment when their final height was reached. Bioimpedance was performed in eight patients; two patients were too heavy for the scale and one refused. The median fat percentage was 33.3 (24.6–55.8), FM 27.1 (19–80.6) kg and FFM 49.8 (41.5–75.8) kg. In one man and one woman, FM was higher than FFM; in the other six patients, FM was >10 kg less than FFM. Compared to the GH-treated group (men and women), no differences in fat percentage, FM, or FFM were seen (*p* = 0.604, *p* = 0.542, and *p* = 0.928, respectively). One man with uncontrolled diabetes suffered from severe heart failure and one man was diagnosed with gastrointestinal cancer.

## 4. Discussion

In this study of 22 adults with PWS who had been treated with GH for a median time of 20 years, all had a higher FFM than FM, although the difference between FFM and FM was less pronounced in women. No major side effects related to GH treatment were noticed. Hormone or metabolic profiles were within normal range, except for a higher fasting glucose and HbA1c in the patients with known diabetes.

One of the major characteristics of PWS is an abnormal body composition, with more body fat than lean body mass [4,5]. GH treatment has anabolic effects and increases lipolysis. In a recent systematic review of GH treatment in adults with PWS, improvement in body composition with an increase in lean body mass and a reduction in fat mass was found in all studies [10]. In another study, improvement in body composition was observed in all patients with PWS, irrespective of their baseline GH status [13]. Our results add to those findings by showing a normal relationship between FM and FFM after 20 years of GH treatment. The typical body composition for PWS with more FM than FFM was only seen in two patients in the non-GH treated group. As expected, fat percentage was higher in women and FFM higher in men [14]. In our patients, GH treatment was initiated in both childhood and adulthood but, as we have shown previously, GH’s positive effect on body composition is independent of the age at which the treatment started [12]. A consequence of the improved body composition would be an increase in physical activity. An improvement in muscle grip strength of 13% was shown in one study [15] and in exercise endurance by 19% in another [16]. In addition, an improvement in peak expiratory flow (PEF) of 12% has been reported [17]. All our patients performed regular physical exercise, but the amount was not recorded. Physical activity is an important determinator for body composition and would be expected to further augment the positive changes in body composition generated by GH. However, it is unclear if the degree of exercise differed between genders.

GH’s antidiuretic effect results in fluid retention. Furthermore, GH and IGF-I have mitogenic and anti-apoptotic properties and might induce tumorigenesis. The most common side effects of GH treatment are transient oedema, muscle pain, and headache, usually related to GH dose [18]. In general, these side effects are related to GH and IGF-I levels, age, and gender, with younger individuals and women needing higher doses than older individuals and men. In the present study, complaints of fluid retention or headache related to GH treatment were not reported, probably because the patients had been on stable GH doses for many years. Of note, GH doses were higher in young individuals, particularly in some of the young women. GH treatment might lead to a decrease in cortisol and thyroxine levels [18]. In this study, morning cortisol and thyroid hormone levels were normal but, as this is not a longitudinal study, we were unable to see if the levels had changed over time.

GH’s physiological effects on the metabolism are of importance in relation to the potential side effects of GH treatment. GH induces insulin resistance by the stimulation of glycogenolysis and lipolysis and the inhibition of glycogenesis and lipogenesis; therefore, GH treatment would be expected to increase blood glucose and improve the lipid profile. Large observational studies of GH-treated adults without PWS have found either no increase or a mild increase in fasting blood glucose and reduced levels of LDL-cholesterol [18]. In accordance, a recent meta-analysis demonstrated that, in studies with a shorter duration (6–12 months), glucose metabolism, including fasting glucose, fasting insulin, and insulin resistance (HOMA-IR), deteriorated. However, these negative effects were not seen with a longer duration of GH treatment, except for a small increase in fasting glucose [19]. Similar findings were reported in a previous meta-analysis of eight GH trials in adults with PWS [20]. The prevalence of diabetes in adults with PWS (without GH treatment) has been reported to be 10–25%, with severe obesity identified as a significant risk factor for developing diabetes type 2 in PWS [21]. Insulin sensitivity is high in PWS, and it might be due to a relatively low amount of visceral fat compared to the total amount of fat, high levels of ghrelin, and impaired GH secretion [21,22,23]. In addition, it has been reported that impaired glucose metabolism during GH treatment in PWS was mainly due to weight gain rather than GH administration [24]. Thus, more important than GH in affecting insulin sensitivity seems to be the variation related to the patient’s genetic predisposition, age, body composition, underlying diseases, and treatments [21,25]. In the present study, fasting glucose and HbA1C did not exceed the normal range, except in the patients with known type 2 diabetes. In line with previous studies, three patients who discontinued GH treatment before this study had increased heavily in weight and developed uncontrolled diabetes. The metabolic control did not improve even though GH treatment was discontinued. Diabetes mellitus is not an absolute contraindication to GH treatment but, in patients with diabetes or in those predisposed to developing diabetes, the close monitoring of glucose metabolism is of great importance [9,21,24]. If type 2 diabetes develops, pharmacological treatment should follow the general guidelines for diabetes [21,25]. A common definition of dyslipidaemia is a total cholesterol > 5.0 mmol/L, LDL > 3.0 mmol/L, HDL < 1.0 mmol/L, and triglycerides > 1.7 mmol/L. According to this definition, none of the patients fulfilled the definition of dyslipidaemia.

Other contraindications to GH treatment are severe obesity, untreated severe obstructive sleep apnoea (OSA), active malignancy, and active psychosis [9]. The risk of cancer in PWS is currently unknown, but cancer as a cause of death is rarely reported [26,27]. Mortality is increased in PWS but generally caused by diseases secondary to obesity, including cardiovascular diseases and respiratory failure [26,27,28]. The median GH dose in the present study was 0.5 mg/day, and the IGF-I levels within the normal age-matched range. This is in accordance with the current consensus guideline, which suggests that the GH dose in adults with PWS should be titrated to an IGF-I level between 0 and +2 SDS for age-matched controls for optimal effects and minimal risk of side effects [9]. IGF-I values at the start of GH treatment for the patients in our study were not available, and it was therefore not possible to quantify the increase in IGF-I over time. However, IGF-I was higher in the GH-treated group compared to the non-GH-treated group. GH deficiency is accepted as a part of PWS [5]. Even though patients with PWS do not fulfil the cut-offs for GH deficiency defined for other patient categories, it must be kept in mind that multisystem conditions may necessitate different interpretations of the GH stimulation test and IGF-I levels from that of the typical population.

For optimal effects of GH treatment, good adherence is important. Daily injections might be inconvenient, painful, or distressing and lead to poor adherence [29]. The poor adherence has been a major incentive for the development of long-acting GH preparations (LAGH), with the hypothesis that fewer injections would increase the adherence and thereby the efficacy [18]. Several different formulations for prolonging the duration of the GH injection for one to two weeks have been developed and are in different study phases or have been registered [30]. LAGH preparations have different pharmacokinetics and pharmacodynamics compared to daily GH, and there may be efficacy, dosing, and safety issues specific to these formulations that are still unsolved, especially for long-term use [18,30]. However, no differences in efficacy or short-term safety have been observed [18,31]. Further studies are needed, but the LAGH preparations have the potential to become an alternative to the currently available GH treatment options, especially in neurodevelopmental diseases such as PWS.

A strength of the present study is the long duration of GH treatment (median 20 years). PWS is a rare, complex, multi-systemic disease and the number of patients in the present study was limited and the patients heterogeneous, thus it cannot be excluded that this might have affected the results. Other limitations are the lack of a sufficiently large control group of non-GH-treated patients for robust comparisons as well as the cross-sectional design of the study. Further, body composition was evaluated with bioimpedance and it can be argued that this might not be the most correct analysis of body composition.

## 5. Conclusions

In conclusion, in this study adults with PWS on treatment with GH for a median time of 20 years had a normal body composition with more FFM than FM. However, GH’s lipolytic and/or anabolic effect was less pronounced in women. No major safety issues related to the GH treatment were recorded. PWS is a complex, multisystemic disease and continuous, individual considerations are required during GH treatment, especially in patients with risk factors for adverse effects. The new LAGH preparations might offer attractive alternatives to optimise GH treatment in PWS.

## Figures and Tables

**Figure 1 jcm-10-02667-f001:**
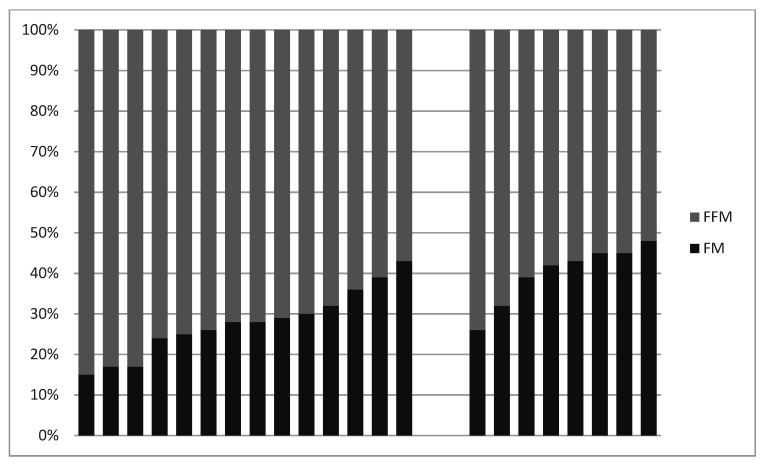
Relationship between percent fat mass (FM) and percent fat free mass (FFM) in 22 adults with PWS, left to right: 14 men on GH treatment, 8 women on GH treatment.

**Table 1 jcm-10-02667-t001:** Laboratory analyte and measurement characteristics of 33 adults with Prader-Willi syndrome. Results are shown as medians and ranges.

	GH Treatment (*n* = 22)	No GH Treatment (*n* = 11)	*p*-Value
**Hormone values**			
IGF-I (µg/L)	208 (85–475)	147 (12–313)	0.016 *
TSH (mU/L)	1.9 (0.7–3.1)	1.5 (0.7–3.2)	0.589
Free T4 (pmol/L)	15 (12–21)	14 (12–16)	0.424
Free T3 (pmol/L)	4.4 (2.9–6.5)	4.9 (3.5–6.0)	0.631
Cortisol (nmol/L	280 (138–618)	337 (157–547)	0.412
**Metabolic values**			
Fasting glucose (mmol/L)	5.4 (4.6–6.4)	11.2 (5.0–19.6)	0.271
HbA1C (mmol/mol)	35 (28–57)	94 (33–109)	0.024 *
Total Cholesterol (mmol/L)	3.8 (2.8–5.8)	4.2 (3.3–5.0)	0.787
LDL-Cholesterol (mmol/L)	2.1 (1.1–3.7)	2.3 (1.7–3.2)	0.660
Triglycerides (mmol/L)	0.87 (0.46–2.20)	1.5 (0.61–1.80)	0.928

* *p* < 0.05. GH—growth hormone; IGF-I—insulin-like growth factor I; TSH—thyroid-stimulating hormone; T4—Thyroxine; T3—Triiodothyronine; HbA1C—glycated haemoglobin; LDL-cholesterol—low-density lipoprotein cholesterol.

**Table 2 jcm-10-02667-t002:** Characteristics by gender of 22 adults with PWS treated with GH for a median of 20 years. Results are shown as medians and ranges.

	Men (*n* = 14)	Women (*n* = 8)	*p*-Value
Age (years)	34 (20–54)	20 (19–52)	0.047 *
BMI (kg/m^2^)	27.0 (25.2–37.5)	29.5 (23.7–36.6)	0.407
Fat (%)	28.1 (14.7–39.3)	42.0 (26.2–48.3)	* p * = 0.0034 *
Fat mass (kg)	24.3 (15.0–47.2)	31.8 (15.9–38.2)	0.124
Lean body mass (kg)	59.2 (45.5–74.3)	42.9 (40.1–51.5)	* p * = 0.0002 *

* *p* < 0.05. BMI = body mass index.

## Data Availability

The data are not publicly available due to privacy and ethical restrictions. The data that support the findings of this study are available on request from the corresponding author.

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
