# Peer review of "Twenty Years of GH Treatment in Adults with Prader-Willi Syndrome"

_jcm, 2021, doi:10.3390/jcm10122667_

Round 1
Reviewer 1 Report
Congratulations to the authors for their good attempt to write this article.
- According to the journal's instructions for authors, "the abstract should be a single paragraph and should follow the style of structured abstracts, but without headings". Therefore, please remove the name of each section from the abstract (e.g., Objective, Study Design, etc.).
- Please consider changing the sentence in the abstract: “In adults, characterized by muscular hypotonia, a different body composition with more body fat than muscle mass, hyperphagia, behavioral problems and cognitive dysfunction”.
- Please consider changing the sentence (line 143): “In this study 22 adults with PWS, who had been treated with GH for a median time of 20 years, all had a higher FFM than FM, although the difference was less pronounced in women”.
- Please specify at what age and after how many years of therapy the patients who developed uncontrolled diabetes discontinued therapy.
- Please consider changing the sentence (line 244): “No major safety issues related to the GH treatment were recorded and uncontrolled diabetes was only seen in the non-GH treated group”. In fact, they developed uncontrolled diabetes before this study.
- Table 1 and 2: I suggest using a supplementary column for p value.
- Table 2: please control brackets.
- Supplementary Table 1: abbreviations should be given in the legend below the table (CV%).
- Supplementary Table 1: I suggest improving layout.
Author Response
Congratulations to the authors for their good attempt to write this article.
We thank the reviewer for this nice comment.
- According to the journal's instructions for authors, "the abstract should be a single paragraph and should follow the style of structured abstracts, but without headings". Therefore, please remove the name of each section from the abstract (e.g., Objective, Study Design, etc.).
Answer: Thank you for this comment. We have now removed the headings.
- Please consider changing the sentence in the abstract: “In adults, characterized by muscular hypotonia, a different body composition with more body fat than muscle mass, hyperphagia, behavioral problems and cognitive dysfunction”.
Answer: Thank you. We have rewritten the sentence and it now reads: “In adults, the syndrome is characterized by muscular hypotonia, a different body composition with more body fat than muscle mass, hyperphagia, behavioral problems and cognitive dysfunction.”
- Please consider changing the sentence (line 143): “In this study 22 adults with PWS, who had been treated with GH for a median time of 20 years, all had a higher FFM than FM, although the difference was less pronounced in women”.
Answer: We have changed the sentence to: “In this study of 22 adults with PWS, who had been treated with GH for a median time of 20 years, all had a higher FFM than FM, although the difference between FFM and FM was less pronounced in women.”.
- Please specify at what age and after how many years of therapy the patients who developed uncontrolled diabetes discontinued therapy.
Answer: Thank you for this suggestion, we have added the following text in the Result section on page 7: One woman developed uncontrolled diabetes at the age of 19 years, one women at the age of 33 years and the man at the age of 34 years. They had all been treated with GH since childhood.”
- Please consider changing the sentence (line 244): “No major safety issues related to the GH treatment were recorded and uncontrolled diabetes was only seen in the non-GH treated group”. In fact, they developed uncontrolled diabetes before this study.
Answer: We thank the reviewer for this comment. The sentence has been amended and the last part removed.
- Table 1 and 2: I suggest using a supplementary column for p value.
Answer: We have now added columns for p-values to the right in the tables
- Table 2: please control brackets.
Answer: The table has been corrected.
- Supplementary Table 1: abbreviations should be given in the legend below the table (CV%).
Answer: We have added explanations for the abbreviations to the legend and put CV% in parentheses.
- Supplementary Table 1: I suggest improving layout.
Answer: The lay-out of the table unfortunately changed when the manuscript was converted to a PDF-file, but we have also made a few changes.
Reviewer 2 Report
The goal of this analysis, presenting the long-term effects of growth hormone treatment in PWS is well-founded and important. As noted by the authors, literature to date has focused on short-term safety rather than longer term risks. In this respect, even though this is a limited number of patients, the longevity of treatment provides positive indicators of the safety of growth hormone in adults with PWS which is of distinct value.
The lack of differences in hormonal and metabolic factors between the treated and untreated groups is of importance, particularly with regard to glucose metabolism. However, there is no data providing direct comparison in FM or FFM between treated and untreated groups. This may be due to insufficient numbers, but the authors should either provide the direct comparison or leave out the untreated group in this portion of the report.
The authors also fail to comment on the fact that IGF-1 level were not significantly different between the two groups and therefore it is unclear why you would expect to see difference in outcomes between the two. They do comment in the discussion that the goal is normal IGF-1, so showing the comparison from a biochemical point of view is helpful, but would not speak to the metabolic advantages of GH treatment. Knowing if the GH treated patients had evidence of GH deficiency (lower IGF-1 pre-treatment, etc.) would be helpful. This data may not be available, but lack of this data again makes the comparison of FM and FFM in these groups less compelling with respect to the positive effects of GH.
The authors make the statement that there is "maintained and improved body composition" in those treated with growth hormone, but there is no basis for this statement as there is no basis for comparison. If the data is available, comparison with prior measurements of BMI, FM and FFM from earlier visits would be a compelling way to show maintenance and improvement of body composition. If no comparison can be made to a control group or with prior data for the individual patients involved, the authors would be better served presenting the data on longevity of treatment and lack of complications/co-morbidity associated with GH therapy.
Overall, the longevity of treatment is novel and the safety data should be reported, but the authors should not over-reach in stating the advantages of GH therapy without some comparison. Perhaps stating just that higher FFM compared to FM would be consistent with prior studies showing these advantages, but leaving it at that.
Author Response
The goal of this analysis, presenting the long-term effects of growth hormone treatment in PWS is well-founded and important. As noted by the authors, literature to date has focused on short-term safety rather than longer term risks. In this respect, even though this is a limited number of patients, the longevity of treatment provides positive indicators of the safety of growth hormone in adults with PWS which is of distinct value.
Answer: We thank the reviewer for the positive comments.
The lack of differences in hormonal and metabolic factors between the treated and untreated groups is of importance, particularly with regard to glucose metabolism. However, there is no data providing direct comparison in FM or FFM between treated and untreated groups. This may be due to insufficient numbers, but the authors should either provide the direct comparison or leave out the untreated group in this portion of the report.
Answer: We thank the reviewer for this important comment. We compared the body composition data from the GH treated group with the data from the non-GH treated group and the following sentence has been included (page 7): Compared to the GH treated group (men and women) no differences in percent fat, FM or FFM were seen (p=0.604, p=0.542 and p=0.928 respectively).” Thus, no differences were seen, and it cannot be excluded that the heterogeneity and a small number of patients in the non-GH treated group affected the results, which is now mentioned in the limitations of the study (page 12). In addition, the following sentence has been added: “The typical body composition for PWS with more FM than FFM was only seen in two patients in the non-GH treated group.”
The authors also fail to comment on the fact that IGF-1 levels were not significantly different between the two groups and therefore it is unclear why you would expect to see difference in outcomes between the two. They do comment in the discussion that the goal is normal IGF-1, so showing the comparison from a biochemical point of view is helpful but would not speak to the metabolic advantages of GH treatment. Knowing if the GH treated patients had evidence of GH deficiency (lower IGF-1 pre-treatment, etc.) would be helpful. This data may not be available, but lack of this data again makes the comparison of FM and FFM in these groups less compelling with respect to the positive effects of GH.
Answer: We thank the reviewer for these important comments. We have re-calculated all statistics and discovered an error in the p-value for the comparison of the IGF-I values between the groups. The GH treated group had higher IGF-I levels and this has now been corrected in the text (page 6) and in Table 1. Unfortunately, we do not have data on IGF-I at start of GH treatment and evaluation of GH deficiency in PWS is difficult. The following text has been included (page 11):” IGF-I values at start of GH treatment for the patients in our study were not available, and it was therefore not possible to quantify the increase in IGF-I over time. However, IGF-I was higher in the GH treated group compared to the non-GH treated group. GH deficiency is accepted to be a part of PWS [5]. Even though patients with PWS do not fulfill cut-offs for GH deficiency defined for other patient categories, it must keep in mind that multisystem conditions may necessitate different interpretation of GH stimulation test and IGF-I levels from that in the typical population.”
The authors make the statement that there is "maintained and improved body composition" in those treated with growth hormone, but there is no basis for this statement as there is no basis for comparison. If the data is available, comparison with prior measurements of BMI, FM and FFM from earlier visits would be a compelling way to show maintenance and improvement of body composition. If no comparison can be made to a control group or with prior data for the individual patients involved, the authors would be better served presenting the data on longevity of treatment and lack of complications/co-morbidity associated with GH therapy.
Answer: We thank the reviewer for the important suggestions. We agree and have put the focus on safety of long-term GH treatment. We prefer to keep the data on the non-GH treated group in the manuscript as we believe that they would be of interest for many of the readers. However, the significance of the findings in this group has been downplayed. Several changes have been made in the manuscript, highlighted in yellow in the manuscript.
Overall, the longevity of treatment is novel, and the safety data should be reported, but the authors should not over-reach in stating the advantages of GH therapy without some comparison. Perhaps stating just that higher FFM compared to FM would be consistent with prior studies showing these advantages but leaving it at that.
Answer: We agree and the manuscript has been revised accordingly. Several changes have been made, highlighted in yellow in the manuscript.
Round 2
Reviewer 2 Report
Thanks to the authors for taking the time to respond to prior comments. The manuscript is much improved with respect to clarity and conclusions.
There are still a few English syntax errors, but these do not significantly impact the content or interpretation of the data.